# Who Will Save the Savior? The Relationship between Therapists’ Secondary Traumatic Stress, Secondary Stress Self-Efficacy, and Attitudes toward Trauma-Informed Care

**DOI:** 10.3390/bs13121012

**Published:** 2023-12-13

**Authors:** Miriam Rivka Miller Itay, Maria Nicoleta Turliuc

**Affiliations:** Faculty of Psychology and Educational Sciences, Alexandru Ioan Cuza University of Iași, 700506 Iaşi, Romania; turliuc@uaic.ro

**Keywords:** preschool children, secondary traumatic stress, secondary traumatic self-efficacy, trauma-informed care

## Abstract

Therapists who treat traumatized preschool children are vulnerable to secondary traumatic stress. This study investigates the relationship between therapists’ attitudes toward trauma-informed care (TIC) and risk of secondary traumatic stress, with secondary traumatic self-efficacy as a mediating variable. Participants included Israeli social workers (*N* = 101) in preschool trauma frameworks, with 97.2% following trauma-informed care principles. The questionnaire combined three instruments: attitudes related to TIC (ARTIC), secondary traumatic stress (STS), and secondary traumatic stress efficacy (STSE). Therapists with less positive attitudes toward trauma-informed care showed higher levels of secondary traumatic stress (*r*[99] = −0.23, *p* = 0.019), while more positive attitudes predicted higher levels of secondary traumatic stress efficacy (*r*[99] = 40, *p* < 0.001). Secondary traumatic self-efficacy mediated the relationship between attitudes toward trauma-informed care and secondary traumatic stress (z = 2.72, *p* = 0.006). Therapists’ secondary traumatic stress may be reduced by improving positive attitudes toward trauma-informed care and enhancing their secondary traumatic self-efficacy.

## 1. Introduction

Young children are at a high risk of being exposed to trauma [1]. They are more vulnerable because their development is accelerated and they have not yet developed sufficient tools and skills to cope, in addition to being dependent on their main caregivers for physical and emotional protection [2,3,4]. Children’s exposure at a young age to traumatic experiences has serious psychological, physiological, developmental, and sociological consequences for their continued development and mental health [2,5,6,7,8]. Since young children have the potential to significantly improve their mental health and well-being, it is possible through early intervention to minimize the effect of their ongoing traumatic experiences [2,3,9,10,11]. Therapy for children who, despite their young age, have experienced extreme traumatic events leaves a mark on everyone around them, and especially on their therapists [12,13,14].

A recommended intervention for this target population is trauma-informed care (TIC) practices [15]. Working with traumatized children requires the therapist’s emotional involvement and may also raise the therapist’s own painful childhood memories and experiences [13]. The therapist’s risk of suffering from secondary traumatic stress (STS) is of critical importance in a TIC setting. General personal safety and, in particular, protection from secondary traumatization are the two key principles in TIC, both for the client and for the therapist [13,16,17] Previous studies have examined the link between attitudes related to TIC and STS more often in the case of educators [18,19]. Therefore, in the present paper, we aim not only to investigate the relationship between therapists’ attitudes toward TIC and their STS, but also to identify one of its important explanatory mechanisms: secondary traumatic self-efficacy.

### 1.1. Attitudes toward Trauma-Informed Care

There is no consistent and absolute consensus on the definition of TIC [20,21]; the terminology in the literature includes parallel terms referring to initiatives, organizations, or systems designed to support people with histories of adverse childhood experiences (ACEs) and trauma. Adverse childhood experiences (ACEs) involve the child’s exposure to traumatic events, including physical, sexual, and emotional abuse, experience of household dysfunction (e.g., divorce/separation, domestic violence), and living with an adult experiencing poor mental health (psychopathology, substance use/misuse, or incarceration), as Felitti et al. indicated [22]. The children’s exposure to adverse experiences may increase the levels of traumatic stress during or after ACE exposure, disrupting healthy brain development in childhood, and may be related to greater risk of poor physical or mental health in adulthood [23].

The basic approach of TIC is based on recognizing and understanding the signs and symptoms of trauma, including profound, extensive physical and psychological effects on survivors’ confidence, and focuses on empowering clients and helping them regain control of their lives. In addition, TIC avoids processes and practices likely to re-traumatize individuals who have experienced trauma in the past and values clients’ involvement in the therapeutic process [17]. Attitudes toward TIC are attitudes that help realize and assimilate the TIC approach, which are a critical factor in the trauma-informed system [20]. The process of change in human services, which takes place when transitioning to TIC, starts with receiving quality information that is effective and relevant, followed by a change in beliefs and attitudes that, in turn, leads to behavioral change [12]. This includes changes in soft skills at an individual level such as education, training, and personal beliefs, and, within an organizational context, organization structures, features, processes, and procedures. The attitudes of therapists toward change, or toward the requirement to adopt a new approach in their work, depend on four conditions: their knowledge and information about the new required approach, personal characteristics that allow or prevent change, openness and general flexibility to change, and the size of the gap between the existing accepted approach and the new approach [24]. TIC training may enrich therapists’ knowledge, and the effectivity of the training in changing therapists’ attitudes may be assessed by analyzing their day-to-day practical work [25,26,27,28] When leading organizations to take practical day-to-day steps to assimilate the TIC approach, it is very important for employees to receive horizontal system support from their colleagues and vertical support from their supervisors and organizational leadership figures [12,25,29,30].

### 1.2. Secondary Traumatic Stress

Secondary traumatic stress (STS) has been defined as stress reactions or symptoms that mirror post-traumatic stress disorder (PTSD), as Christian-Brandt and her colleagues underlined [18]. If PTSD was initially recognized only for individuals directly exposed to trauma, STS appears only as a result of indirect exposure to trauma [31,32]. It represents the emotional distress that those who help trauma victims undergo, which may lead them to display symptoms such as avoidance behaviors, psychological arousal, feelings of sadness, inability to function, and so on Figley [31]. Social workers and therapists are at a higher risk of suffering from STS because of their stressful working conditions and because they are more exposed to trauma in their routine work, especially if they provide welfare services to children with a history of trauma [12,14,33] Studies show that social workers who treat traumatized children are at risk of suffering from STS, and even more so if they themselves have experienced trauma in their childhood [13,33,34]. STS in therapists affects the well-being of clients and the quality of their service [12,35] since STS may cause mistakes in diagnoses, professional decisions, and therapy planning, as well as intolerance toward clients, and may also create stressful interpersonal relationships and unsatisfactory clinical results. Therapists suffering from STS are also likely to be absent from work more often [30,36,37]. When a therapist has a high level of STS, they are in a survival mode, which significantly influences the nature and quality of the services that they provide and, the performance and the scope of tasks carried out for clients who suffer from trauma [38]. Garwood et al. [12] found that improvement in TIC-related knowledge associated with lower levels of STS and increased well-being is more likely to exist among therapists with more experience and who are older in age than among younger team members. Experienced therapists were shown to have lower STS levels due to their accumulated life experience and professional experience in dealing with clients with a history of trauma [12,39,40]. The TIC model includes tools that may be used by professionals for their own well-being, as well as signs for recognizing STS in therapists [13,16,41,42]. Research suggests ways to prevent STS, such as offering trauma-informed peer support, as well as strengthening and supporting organizational structures that increase the personal resilience of therapists and reduce the risk of suffering from STS. Moreover, the support of colleagues and supervisors has an effect on organizational atmospheres and cultures since colleagues and supervisors can work together to help reduce STS [13,38,43,44,45]. Organizational interventions to prevent STS must focus on developing a balanced work–home system, as well as on establishing coping competences and strategies in therapists, especially therapists working in child prevention services, who are at an increased risk of developing STS [39].

### 1.3. Secondary Traumatic Stress Efficacy

According to social cognitive theory, self-efficacy is a sense of control over one’s environment, as well as a perceived ability to deal with challenges and demands (such as major stressful events and their consequences) using adaptive actions [46]. Self-efficacy is attained and expressed differently in each individual, due to differences in feelings, thoughts, and reactions [46]. Social cognitive theory contends that people’s belief in their ability to cope, which stems from their inherent self-efficacy, helps them to overcome difficulties, including those that result from exposure to traumatic experiences [47]. Research has focused on self-efficacy in the context of the workplace, including how it may be used in training human service professionals to efficiently handle tasks and challenges that come up in their work [48,49].

Cieslak et al. [50] suggested using this advantage and developing self-efficacy as a tool for therapists to protect themselves from the effects of trauma when treating at-risk post-trauma clients. The research measured the effect of self-efficacy on therapists who worked in very stressful jobs and whose quality of life was consequently compromised. The researchers assumed that therapists with higher levels of secondary traumatic self-efficacy (STSE) had a higher quality of life, less compassion fatigue, and less STS. On top of that was measured secondary traumatic growth, which is defined as positive change in life due to indirect exposure to trauma. The results show that therapists with a strong belief in their ability to deal with clients with a history of trauma result in lower STS levels. Cieslak et al. [50] It was also found that STSE increases when there is greater social support from family, friends, and other significant sources [48,50,51].

Therefore, it is important to explore whether attitudes toward trauma-informed care can be a predictor of therapists’ secondary traumatic stress, and whether therapists’ secondary traumatic self-efficacy is a mediator of this relationship. In this case, positive attitudes related to trauma-informed care and higher secondary traumatic stress efficacy could be a multiplier of strength for therapists to reduce the risk of secondary traumatic stress.

### 1.4. The Current Study

The aim of the current study is to investigate the relationship between attitudes toward TIC and secondary traumatic stress (total score and its dimensions) in therapists working with children with a history of trauma, and the explanatory mechanism of this relation. Thus, we explored five hypotheses:

**H1.** 
*The secondary traumatic self-efficacy mediates the relationship between attitudes related to TIC and secondary traumatic stress (total score).*


**H2.** 
*The therapist’s attitude toward TIC and their demographic and professional characteristics (age, years of seniority, education level, TIC training) are predictors of secondary traumatic stress.*


**H3.** 
*The therapist’s attitude toward TIC and their demographic and professional characteristics (TIC training and therapy setting) are predictors of secondary traumatic self-efficacy.*


**H4.** 
*The therapist’s secondary traumatic self-efficacy and their demographic and professional characteristics (age, years of seniority, education level, TIC training) are predictors of secondary traumatic stress.*


**H5.** 
*The therapist’s attitude toward TIC, their secondary traumatic self-efficacy, and demographic and professional characteristics (age, years of seniority, education level, TIC training) are predictors of secondary traumatic stress.*


## 2. Methods

### 2.1. Participants

The study participants were therapists (*N* = 101) working with traumatized preschool children, aged between 1 and 6 years, and their parents in Israel. Many of the participating therapists had already received TIC training from the Ministry of Welfare and Social Security, in partnership with the particular municipality and non-governmental organization (NGO) that administered the service/center for which the therapists worked. Most of the funding and TIC supervision was provided by the Israeli government, from the Ministry of Welfare and Social Security budget (see Appendix A). Most of the therapists who participated in the study were Jewish ultra-orthodox (*Haredi*) or national religious (72.3%) and the others were secular. The educational level of the participants was high, with 88.1% holding an M.A or Ph.D. degree. The majority of participants were female (89%) with a small number of males (11%). Almost half of the therapists had undergone intensive TIC training, and almost half indicated that their TIC training was initiated by their workplace. Over 60% of the therapists worked in local municipal service centers that were not located in security emergency areas. Those locations were not vulnerable to rocket attacks. More than half of the therapists indicated that they had TIC supervision (58.4%) and that their clients were orthodox or religious (65.4%). Finally, most of the therapists indicated that they did not work in a dyad setting.

For the purpose of the study, three questionnaires, detailed below, were combined, together with a list of additional background questions, and distributed to a wide variety of preschool children’s therapists from various centers throughout Israel. The questions referred to the therapists’ main place of work. A convenience sample was used. The sample size provided the main analyses with sufficient statistical power (0.90). For conducting linear multiple regression, a fixed model was used, with an R2 increase using the test parameters (low-medium effect size = 0.15, α error = 0.05, power = 0.90, number of tested predictors = 2—the ARTIC and STSE scores—and total number of predictors = 22: 20 demographic and professional background characteristics of therapists and 2 tested predictors); the total sample size required was at least 89 participants.

### 2.2. Measures

The questionnaire used in the study consisted of the following three self-report scales.

#### 2.2.1. Attitudes Related to the TIC (ARTIC) Scale [20] 

The Human Services 45-item version of this questionnaire was used to measure the extent to which professionals who provide social services, personally or at educational institutions, to people with a history of trauma were themselves trauma-informed. This version of the questionnaire measures five core areas: the underlying causes of problem behavior and symptoms, responses to problem behavior and symptoms, on-the-job behavior, SE at work, and reactions to the work. The questionnaire also measures two supplementary areas: personal support and system-wide support of TIC. In the present study, we used the total score of ARTIC, and the score was calculated by averaging the items, with a higher score indicating a more favorable attitude toward trauma-informed care. The internal consistency of Cronbach’s alpha in the current study was very high: α = 0.90.

#### 2.2.2. Secondary Traumatic Stress (STS) Questionnaire [36]

This questionnaire measures the STS of professionals who are indirectly exposed to trauma due to working with traumatized clients. The questionnaire includes 17 items, which measure post-traumatic stress disorder according to the criteria that appear in the *Diagnostic and Statistical Manual of Mental Disorders* (DSM-IV): intrusion, avoidance, and arousal. The internal consistency of Cronbach’s alpha for the total score of the questionnaire was very high, with α = 0.93. The reliability alpha level for the three factors—intrusion, avoidance, and arousal—were high (α = 0.78, α = 0.86, and α = 0.79, respectively).

#### 2.2.3. Secondary Traumatic Self-Efficacy (STSE) Questionnaire [50] 

This questionnaire measures the ability of professionals to deal with challenges at work, as well as their risk of developing secondary traumatization after being indirectly exposed to trauma from clients. The instrument measures STSE as a protective factor. The internal consistency of Cronbach’s alpha for all the questionnaire items is high: 0.79.

#### 2.2.4. Demographic and Professional Background of Therapists

The independent variables were age, gender, religiosity of the therapists and clients, professional expertise, level of education, level of TIC training, initiator of TIC training, the existence of supervision from a senior therapist who is experienced in TIC, years of seniority as a therapist, and whether or not the therapist works a security emergency area.

### 2.3. Procedure

For the purpose of the study, three questionnaires, detailed above, were combined together with a list of additional background questions, and distributed to a wide variety of preschool children ’s therapists, from various centers throughout Israel. The questions referred to the therapists’ main place of work.

The questionnaire was distributed in two different technological versions—one for smartphones and one for computers. The questionnaire was sent via WhatsApp and email in the form of a link, and was also distributed as a public online form on the community platforms of the Israeli Ministry of Welfare and Social Security and on Facebook, such as to graduates of the CPP (Child Parent Psychotherapy) training program. The study was approved by the ethics committee of Alexandru Ioan Cuza University of Iași, as well as by the research and ethics committee of the Ministry of Welfare and Social Security in Israel. After receiving their approval to conduct the study, the process of distributing the questionnaires to the participants began. Each scale was translated into Hebrew, the native language of the participants.

### 2.4. Data Analysis

Firstly, descriptive and correlational statistical analyses were performed using the software SPSS 28.0.1.0. The mean scores, standard deviations, and Pearson correlation coefficients were calculated for each of the variables. Secondly, four mediation models were proposed and analyzed using Model 4 [52] from PROCESS version 4.0 with IBM SPSS 28. 5000 bootstrap samples used, utilizing confidence intervals based on bootstrapping in order to estimate confidence intervals of 95% [53]. Confidence intervals which do not include zero indicate significant effects [54]. A mediation analysis was conducted to verify the mediating role of secondary traumatic self-efficacy in the relationship between attitudes related to TIC and secondary traumatic stress, total score, and its dimensions.

## 3. Results

### 3.1. Descriptive Statistics and Correlational Analyses

The descriptive statistics and the correlational analyses are shown in Table 1. In order to examine the correlations between the study variables, Pearson correlation analyses were conducted. A significant negative correlation was found between the attitudes toward TIC and secondary traumatic stress (*r*[99] = −0.23, *p* = 0.019). In addition, a significant positive correlation was found between the attitudes toward trauma-informed care and secondary traumatic self-efficacy (*r*[99] = 40, *p* < 0.001). Finally, significant negative correlations were found between the secondary traumatic stress, total score and its various sub-scales, and secondary traumatic self-efficacy (with *r*[99] ranging between −0.25 and −0.45). These results indicate that more positive attitudes toward TIC and higher levels of secondary traumatic self-efficacy are related to the lower levels of secondary traumatic stress of therapists working with children with a history of trauma.

### 3.2. The Mediational Role of Therapists’ STSE in the Relationship between the Attitude toward TIC and STS (Total Score and Dimensions)

To examine whether the STSE serves as a mediating variable between the attitudes related to TIC and the STS (total score), a mediation analysis was conducted using the PROCESS software (Hayes, 2013). The analysis reveals that the relationship between attitudes toward TIC and STS was mediated by the STSE, and the mediation is significant (*z* = 2.72, *p* = 0.006), thus confirming the first hypothesis. Both attitudes related to TIC (*β* = −0.238, *p* = 0.000) and STSE (*β* = −0.371, *p* = 0.000) are significant predictors of STS. The indirect (ab), total (c), and direct (c’) effects were also significant (see Figure 1).

After examining the mediational role of STSE, we have additionally conducted two hierarchical analyses in order to investigate our second and third hypotheses regarding the contribution of the therapists’ demographic and professional characteristics and the unique contribution of the score on the attitudes toward TIC to the EPV of the STS and STSE. The therapists’ demographic and professional characteristics were entered in the first step of the regression models in a stepwise manner. The order of the variables entered was in accordance with their level of significance. In the second step, the score on the attitudes toward TIC was entered. This variable was entered only in the second step of the regression model to examine its unique contribution to the EPV of the STS and STSE, beyond the contribution of the therapists’ demographic and professional characteristics (see Table 2).

*Therapists’ demographic and professional characteristics on STS:* Table 2 shows the four demographic and professional characteristics of therapists that were entered in the first step of the regression model: age, TIC training, years of seniority as a therapist, and level of education. These four demographic and professional characteristics contributed significantly (22.9%) to the EPV of the STS. The β coefficients for “Years of seniority as a therapist” and “Level of educational” are positive, and the β coefficients for “Age” and “TIC training” are negative. These results indicate that therapists with no TIC training, who are younger in age, who are more educated, and who have greater seniority as therapists tend to have higher levels of STS.

*Therapists’ demographic and professional characteristics on STSE:* Table 2 shows the two professional characteristics of therapists that were entered in the first step of the regression model: (1) whether the therapist underwent TIC training and (2) the therapeutic setting. These two professional characteristics contributed significantly (15.4%) to the EPV of the STSE. The positive β coefficients of the two professional characteristics indicate that therapists who undergo TIC training and whose therapeutic setting is dyad tend to have higher levels of SE.

In the second step of both regression models, the attitudes toward TIC contributed significantly to the EPV of the STS and STSE scores (3.9% and 10.8%, respectively), beyond the contribution of the therapists’ professional characteristics. The negative β coefficient indicate that as the positive attitudes toward TIC decrease, the level of STS of therapists who work with children with a history of trauma increases, beyond the influence of the therapists’ professional characteristics. The positive β coefficient indicates that, as the positive attitudes toward TIC increase, the level of STSE of therapists who work with children with a history of trauma increases, beyond the influence of the therapists’ professional characteristics.

An additional two hierarchical regression analysis was conducted in order to examine the fourth and the fifth research hypotheses. The first analysis examined the unique contribution of the STSE and the second analysis examined the unique contribution of the attitudes toward TIC and STSE to the EPV of the STS, beyond the therapists’ demographic and professional characteristics. In the regression analysis, the therapists’ demographic and professional characteristics were entered in the first step of the regression model in a stepwise manner. In the second step of the regression model, the STSE (in the first analysis) and the attitudes toward TIC (in the second analysis) were entered. These variables were entered only in the second step of the regression model to examine their unique contribution to the EPV of the STS, beyond the contribution of the therapists’ demographic and professional characteristics. Finally, in the second analysis, the STSE was entered in the third step of the regression model. This variable was entered only in the third step of the regression model to examine its unique contribution to the EPV of the STS, beyond the contribution of the therapists’ demographic and professional characteristics and the attitudes toward TIC (see Table 3).

Unique contribution of STSE on STS: Table 3 shows that the STSE contributed significantly (11%) to the EPV of the STS score, beyond the contribution of the therapists’ professional characteristics, with a negative β coefficient. This result indicates that therapists with lower SE have higher levels of the STS, beyond the influence of their professional characteristics.

*Unique contribution of attitudes toward TIC and STSE on STS*: Table 3 shows that in the second step, the attitudes toward TIC contributed significantly (3.9%, *p* = 0.028) to the EPV of the STS, beyond the contribution of the therapists’ professional characteristics, with a negative β coefficient. This result indicates that, as positive attitudes toward TIC decrease, the level of STS of therapists who work with children with a history of trauma increases, beyond the influence of the therapists’ professional characteristics. In the third step, the STSE contributed significantly (7.9%) to the EPV of the STS, beyond the contribution of the therapists’ professional characteristics and of the attitudes toward TIC, with a negative β coefficient. This result indicates that therapists with a lower SE have higher levels of the STS, beyond the influence of their professional characteristics and their attitudes toward TIC. In the third step of the regression model, the significance level of the contribution of the attitudes toward TIC decreased and did not reach a level of significance (*p* = 0.292).

## 4. Discussion

The main systemic concern in the study is how therapists working at therapy centers for preschool children can be influenced to hold more positive attitudes toward TIC and attain a higher level of STSE, which will improve their resilience to STS. The current study discusses these issues in depth.

Attitudes evolve and change from a substrate of knowledge, emotion, intentions, and behavior, which are factors that nourish each other. The study examines therapists’ attitudes toward TIC, the relationship between these attitudes and STS, and one of its possible explanatory mechanisms, the STSE, based on the existing literature [20,36,47,50]. The study also creates a content environment for trauma therapists as professionals and as people. The risk that therapists will suffer from STS symptoms increases when they are engaged in the professional field of treating helpless preschool children [20,31,36,45,55]. The study shows that therapists with a less positive attitudes toward TIC have higher levels of STS. Therefore, it is important to increase the positive attitudes of therapists toward TIC to protect them from STS. Moreover, the more therapists are familiar with, receive knowledge about, and are willing to implement the TIC principles in their daily practice, the more positive their attitude is likely to be toward TIC and the less likely they are to suffer from STS. This is expected to decrease the harm to the well-being of therapists, allow them to provide higher-quality service to clients, and create a more positive work environment for their colleagues and organization [30,35,36,37].

In addition, the current study examines the relationship between STSE and STS. In the context of the workplace, SE is the worker’s belief that they are adequately trained for the demands of their job and that they are able to perform their duties in an efficient, effective manner and meet the challenges that they encounter at work [49]. STSE was also found to be a potential protective factor among therapists who work with clients who have a history of trauma [50]. The results indicate that therapists with lower levels of STSE and a less positive attitude toward TIC have higher levels of STS. This leads to the conclusion that the higher the STSE of therapists who work with preschool children with a history of trauma, the lower the risk that the therapists will suffer from STS symptoms, since their SE serves as a protective factor.

Another insight is that therapists with a more positive attitudes toward TIC have higher levels of STSE relative to their average level. To increase therapists’ positive attitudes toward TIC, and to increase their STSE, therapists must first be provided with extensive in-depth information regarding elements of their practice that require change. The next step is to apply this change behavioristically [24].

In the professional therapeutic context, to lead therapists to a change in attitude and implementing their new attitude in their work as therapists with a higher level of SE, they must be provided with work experience and training, as well as supervision in performing tasks and overcoming challenges at work [49,56]. The climate of a “constantly learning organization,” such as is achieved when the workplace initiates training programs [43], allows for the required openness, supports a change in attitude, and promotes high levels of SE. The information and positive experiences shared by colleagues, and especially the attitudes of staff and employees who work together at the same organization, promote a positive change in attitude, which leads to higher levels of SE [24,48]. This provides trauma therapists with significant support within their organization [45,57]. Natural support systems such as family, friends, and other sources of support outside of the workplace were also found to be significant in increasing STSE [50].

The current study shows that therapists who work in a dyad child–parent setting and have received TIC training have high levels of SE. Over the past 15 years, significant steps have been taken to gradually transition to the TIC approach across Israel. The Israeli Haruv Institute [58], founded by the American Schusterman Foundation, brought the child–parent psychotherapy (CPP) intervention [10] method to Israel and initiated intensive continuous training programs for implementing the method. CPP training is an evidence-based practice based on a dyad therapy setting and TIC principles, and produces therapists with high levels of SE [56]. Using evidence-based practice intervention was found to be another factor that prevents STS and leads to compassion satisfaction, which is the pleasure that comes with effective work [39]. The questionnaire used in the current study was distributed to graduates of the CPP training program, all of whom worked in a dyad setting, and the results are in accordance with the findings of David and Schiff [56] concerning the high levels of SE in CPP therapists.

The Israeli Ministry of Welfare and Social Security has also initiated training programs based on the dyad child–parent therapy setting and TIC principles, in recent years, for therapists who work with preschool children and their parents. These programs include training for parenting therapy and attachment-oriented therapy, and are supported by local authorities and NGOs [41]. The programs include courses of various intensities and lengths, some of which are adapted to and held in the workplace, and some of which are initiated by the State of Israel and run by a national training center.

The main result of the current study is the significant negative association between attitudes toward TIC and STS, a relationship mediated significantly by STSE. This result is very important for practical purposes and demonstrates the significance of STSE as a mediator in the relationship between therapists’ attitudes toward TIC and their risk of suffering from STS. In David and Schiff’s [56] study on the implementation of the CPP therapy method in Israel, they propose a training program that includes a broad professional envelope with a social network of participants and ongoing regular professional training. As in the current study, the researchers found that SE serves as a mediator.

TIC training serves to improve knowledge, reduce STS symptoms, and increase therapists’ well-being [12,19]. In the current study, therapists with no TIC training or who were younger in age, more educated, and with greater seniority as therapists tended to have higher levels of STS (STS scale). The absence of TIC training led to more negative attitudes toward TIC and higher levels of STS, and SE cannot be explained to serve as a protective factor for therapists who lack TIC training. This can be understood in the context of trauma therapy in Israel. For several decades now, there have been a number of trends in Israel that are reflected among the participants in the current study. The first trend is the direct M.A. program for social workers, which is required as a condition for obtaining a license to work in therapeutic fields, such as for professionals in psychology and art therapy. As a result, there are more young social workers in the field with a higher level of education. Another trend is that of social workers who choose to become professional therapists and thereby attain higher professional status. Since there has been a shortage of therapists in the field of welfare for several years now, these young therapists come prepared academically for the job, and they enter the market sooner than in the past. The consequences are the risk of STS in this group of participants, as found in a study by Garwood et al [12]. Therefore, TIC-specific training must be planned and offered to all therapists and mediators in the various programs for preschool children and their parents with a history of trauma. Mediators in cases of children who suffer from neglect, which is also considered to be a type of child abuse, may also benefit from these training programs. In summary, by receiving adequate training for TIC, gaining knowledge and tools, and developing efficient effective skills, the attitudes of therapists toward TIC improves. In addition, there is a high chance that the therapists’ SE will increase as a result of this process and, hence, their risk of suffering from STS will be significantly reduced. These therapists, who have fewer trauma-related beliefs about themselves and the world, may grow professionally, and their well-being may improve. They may rise to a professional level to the point of secondary traumatic growth [30,45,50,51], find deep broad meaning in their role as trauma therapists for preschool children and their parents, and understand their great contribution in this unique role. They are likely to choose to continue providing benevolent care in the long term, while maintaining self-preservation [51] and professional development.

Beyond its strengths, the present study has several limitations. We used a correctional perspective that does not allow us to infer causal relationships among the variables. Also, we used self-reporting measures that were very lengthy, and this feature might have influenced the validity of the answers. Furthermore, the content of the questionnaire addressed both personal and professional issues. This might have caused distress among the participants, and led them to answer in a less-than-honest manner. Unfortunately, the STS scale was not updated or replaced with another one based the new developments of PTSD in DSM V. Therefore, researchers are still using the version developed by Bride et al. [36] based on DSM IV as we see in Kalaitzaki et al. [59]. Finally, there are concerns regarding the composition of the participants. There was little diversity of religiosity and culture, and very few participants from emergency security areas or from local authorities in which there are no therapeutic services for preschool children. In addition, the participants work only with young children. Including therapists who work with other ages would have added further depth to the study.

## 5. Conclusions

The importance of protecting therapists from STS has been discussed in the past as one of the principles of a trauma-informed approach. The current study discusses the inner world of therapists who treat children with a history of trauma and their parents, since they are the most likely of all types of therapists to suffer from STS. The study shows that therapists’ positive attitudes toward TIC is one protective factor from high STS total scores and accompanying symptoms. In addition, the study shows that STSE is a key mediator between the development of positive attitudes toward TIC and the risk of suffering from STS in trauma therapists who treat preschool children and their parents. An important contribution of this work is the emphasis on the unique population of therapists with high levels of STS, characterized by young age and seniority as therapists in the trauma field, and with no TIC training for treating preschool children. Since there is a lack of therapists and young therapists receive professional opportunities earlier than in the past, it is important to prioritize young therapists in initiatives and budgets for TIC training at the government level. They must be trained before or immediately after starting their job so that they are not negatively affected by their work. To protect therapists from STS, it is important to increase their SRSE by providing widespread TIC and evidence-based practice intervention training programs for all preschool children therapists and continuing high-quality supervision. This must be undertaken at the governmental, municipal, and organizational levels. Some steps to achieve this are already being taken in Israel. Since TIC in Israel has been integrated in a gradual manner, further research is needed to see the changes that a TIC approach will bring to the field for all ages of children with a history of trauma and their parents.

## Figures and Tables

**Figure 1 behavsci-13-01012-f001:**
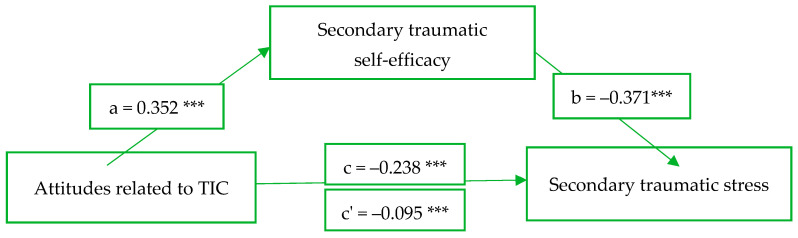
Secondary traumatic self-efficacy as mediator of the relationship between attitudes related to TIC and secondary traumatic stress. *** represents *p* < 0.001.

**Table 1 behavsci-13-01012-t001:** Pearson correlations between attitudes related to trauma-informed care and secondary traumatic stress.

	Descriptive Statistics			Pearson Correlation Coefficients				
Questionnaire Scales and Sub-Scales	M	SD	Range	2	3	4	5	6
** *ARTIC (1)* **	5.63	0.54	4.04–6.76	−0.23 ***	−0.16 **	−0.27 ***	−0.21 **	0.40 ***
** *STS—Total (2)* **	2.18	0.72	1.06–4.94		0.91 ***	0.94 ***	0.93 ***	−0.40 ***
Intrusion (3)	2.38	0.79	1.00–5.00			0.76 ***	0.80 ***	−0.25 *
Avoidance (4)	2.07	0.79	1.00–5.00				0.82 ***	−0.45 ***
Arousal (5)	2.12	0.74	1.00–4.80					−0.38 ***
** *STSE (6)* **	5.67	0.75	2.14–7.00					1

* *p* < 0.05, ** *p* < 0.01, *** *p* < 0.001. Note: ARTIC = attitudes related to trauma-informed care, STS = secondary traumatic stress, STSE = secondary traumatic self-efficacy.

**Table 2 behavsci-13-01012-t002:** Results of two hierarchical regressions: attitudes toward TIC (ARTIC) on secondary traumatic stress (STS) and attitudes toward TIC (ARTIC) on secondary traumatic self-efficacy (STSE), controlling for the therapist’s demographic and professional characteristics.

Steps	Explanatory Variables	B	SE.B	β	*R* ^2^	∆*R*^2^
Unique contribution of the ARTIC to STS						
1	Therapists’ age	−0.02	0.01	−0.30 **	0.089 **	0.089 **
	Therapists’ age	−0.02	0.01	−0.29 **		
	Underwent TIC training ^1^	−0.90	0.34	−0.25 **	0.150 ***	0.061 **
	Therapists’ age	−0.04	0.01	−0.53 ***		
	Underwent TIC training ^1^	−0.86	0.33	−0.24 *		
	Years of seniority as a therapist	0.02	0.01	0.32 *	0.195 ***	0.045 *
	Therapists’ age	−0.04	0.01	−0.54 ***		
	Underwent TIC training ^1^	−0.83	0.33	−0.23 *		
	Years of seniority as a therapist	0.02	0.01	0.29 *		
	Education	0.35	0.17	0.19 *	0.229 ***	0.034 *
2	Therapists’ age	−0.03	0.01	−0.50 ***		
	Underwent TIC training ^1^	0.70	0.33	0.19 *		
	Years of seniority as a therapist	0.02	0.01	0.24		
	Education	0.39	0.17	0.21 *		
	ARTIC	−0.27	0.12	−0.20 *	0.268 ***	0.039 *
Unique contribution of the ARTIC to STSE						
1	Underwent TIC training ^1^	1.29	0.36	0.34 ***	0.115 ***	0.115 ***
	Underwent TIC training ^1^	1.28	0.36	0.34 ***		
	Therapy setting ^2^	0.29	0.14	0.20 *	0.154 ***	0.039 *
2	Underwent TIC training ^1^	1.06	0.34	0.28 **		
	Therapy setting ^2^	0.27	0.13	0.18 *		
	ARTIC	0.46	0.12	0.33 ***	0.262 ***	0.108 ***

* *p* < 0.05, ** *p* < 0.01, *** *p* < 0.001; ^1^ Underwent TIC courses (0 = No, 1 = Yes); ^2^ therapy setting (0 = not dyad, 1 = dyad treatment).

**Table 3 behavsci-13-01012-t003:** Results of two hierarchical regressions: secondary traumatic self-efficacy (STSE) on secondary traumatic stress (STS) and attitudes toward TIC (ARTIC) and STSE on STS controlling for the therapist’s demographic and professional characteristics.

Steps	Explanatory Variables	B	SE.B	β	*R* ^2^	∆*R*^2^
Unique contribution of the STSE to STS						
1	Therapists’ age	−0.02	0.01	−0.30 **	0.089 **	0.089 **
	Therapists’ age	−0.02	0.01	−0.29 **		
	Underwent TIC training ^1^	−0.90	0.34	−0.25 **	0.150 ***	0.061 **
	Therapists’ age	−0.04	0.01	−0.53 ***		
	Underwent TIC training ^1^	−0.86	0.33	−0.24 *		
	Years of seniority as a therapist	0.02	0.01	0.32 *	0.195 ***	0.045 *
	Therapists’ age	−0.04	0.01	−0.54 ***		
	Underwent TIC training ^1^	−0.83	0.33	−0.23 *		
	Years of seniority as a therapist	0.02	0.01	0.29 *		
	Education	0.35	0.17	0.19 *	0.229 ***	0.034 *
2	Age	−0.03	0.01	−0.46 ***		
	Underwent TIC training ^1^	0.39	0.32	0.11		
	Years of seniority as a therapist	0.02	0.01	0.23		
	Education	0.44	0.16	0.23 **		
	STSE	−0.34	0.09	−0.36 ***	0.338 ***	0.110 ***
Unique contribution of the ARTIC and STSE to STS						
1	Age	−0.02	0.01	−0.30 **	0.089 **	0.089 **
	Age	−0.02	0.01	−0.29 **		
	Underwent TIC training ^1^	−0.90	0.34	−0.25 **	0.150 ***	0.061 **
	Age	−0.04	0.01	−0.53 ***		
	Underwent TIC training ^1^	−0.86	0.33	−0.24 *		
	Years of seniority as a therapist	0.02	0.01	0.32 *	0.195 ***	0.045 *
	Age	−0.04	0.01	−0.54 ***		
	Underwent TIC training ^1^	−0.83	0.33	−0.23 *		
	Years of seniority as a therapist	0.02	0.01	0.29 *		
	Education	0.35	0.17	0.19 *	0.229 ***	0.034 *
2	Age	−0.03	0.01	−0.50 ***		
	Underwent TIC training ^1^	−0.70	0.32	−0.19 *		
	Years of seniority as a therapist	0.02	0.01	0.24 *		
	Education	0.39	0.17	0.21 *		
	ARTIC	−0.27	0.12	−0.20 *	0.268 ***	0.039 *
3	Age	−0.03	0.01	−0.45 ***		
	Underwent TIC training ^1^	−0.37	0.32	−0.10		
	Years of seniority as a therapist	0.02	0.01	0.22 *		
	Education	0.44	0.16	0.24 **		
	ARTIC	−0.13	0.12	−0.10		
	STSE	−31	0.09	−0.32 ***	0.346 ***	0.079 ***

* *p* < 0.05, ** *p* < 0.01, *** *p* < 0.001; ^1^ Underwent TIC training; 0 = Did not undergo TIC training, 1 = Underwent TIC training.

## Data Availability

Data that support the findings of this study are available on request from the corresponding author due to privacy issues.

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
