# Peer review of "Who Will Save the Savior? The Relationship between Therapists’ Secondary Traumatic Stress, Secondary Stress Self-Efficacy, and Attitudes toward Trauma-Informed Care"

_behavsci, 2023, doi:10.3390/bs13121012_

Round 1

Reviewer 1 Report

Comments and Suggestions for Authors

Thanks to the authors for sharing the manuscript. I think their research is interesting and promising, but I am concerned about several points:

1.       In the abstract and further in the manuscript, I would recommend that the authors emphasize mediation in the relationship between variables, and not just about mediation between variables. For example, “Secondary traumatic self-efficacy mediated <the relationship> between attitudes towards Trauma-Informed Care and secondary traumatic stress”.

2.       I am confused by the formulations of research hypotheses. Firstly, I suggest removing abbreviations from them, because it will be difficult for the reader to understand the essence of the hypothesis, in which there are three abbreviations. Secondly, I would replace "are predictors" with "will predict".

3.       Do I understand correctly that the participants filled out the measures in Hebrew? If so, the authors should either refer to the adapted versions of the measures, or give a psychometric justification of the measures used. Is it enough to specify Cronbach's alpha to say that the translated instrument is adequate?

4.       The study sample is complex, so its small size is understandable. However, the authors used mediation analysis and regression analysis, and did not show a preliminary calculation of the power analysis. Was it possible to use these statistical procedures for a given sample size?

5.       The Procedure follows the Measures, but in the Procedure the authors write about “three questionnaires, detailed below”. Accordingly, it is necessary to replace “described below” with “described above”.

6.       The results of the mediation analysis are not fully described, I recommend presenting them in the form of a figure so that all the values of the mediation model are visible. In addition, the authors do not justify why they first used mediation analysis and then conducted a series of regression analyses.

It is better to expand the limitations of the study, supplement them with perspectives and transfer them from the Conclusion to the Discussion.

Reviewer 2 Report

Comments and Suggestions for Authors

This article, Who Will Save the Savior? The Relationship Between Therapists' Secondary Traumatic Stress, Secondary Stress Self-Efficacy, and Attitudes Toward Trauma-Informed Care, contains important information that will add to the literature on therapists' secondary traumatic stress.  It is important because STS is a real phenomenon affecting good and talented therapists.

Some recommended edits to the manuscript:

1.  page 2 line 56: the first time you refer to ACES, write it out.  Then give a description. Not everyone will now what ACES means.

2.  page 2 lines 80-82. In your literature review, you are providing background on the variables. You have a section about the measures. Don't include the fact that you measured TIC with the ARTIC here.

3.  page 3 line 125.  You have just introduced self-efficacy in the first line.  You need a reference following the first line. Maybe that reference is (Bandura, 1977).  If so, it should go first after the first line.

4.  I'm confused why you utilized a PTSD measure that measures PTSD from the DSM-IV. The DSM-5 has changed the way that PTSD is evaluated, increasing the number of symptom groups and the number of symptoms. This is a limitation that should be addressed in a section after the discussion.  Do you think that increases or decreases the level of PTSD in your therapists?

5.  When talking about the results, you tended to refer to the scales rather than the concepts. I would love to see that changed. It becomes confusing. There are places where you include the variables in addition to the measures. I don't think you need to include the measures here. You have already addressed the measures in the measures section.  

6.  Your results are interesting and important. It would be helpful to include a figure showing the mediation model 4 (Hayes) that includes X, Y, M, a1, b1, c', c.

7.  You've done a nice job tying the results to the conclusion - need to keep learning which is a protective factor for therapists and the development of STS. I liked too how you tied that conclusion to the way in which they train therapists in Israel. Nice work! 8. In your conclusion, include the problem with your PTSD measure.

Comments on the Quality of English Language

1.  Abstract line 9: This study investigates (a study cannot investigate).  Instead... We investigated the relationship between....

2.  page 2 line 62: Attitudes toward TIC are attitudes that help realize and assimilate the TIC approach. Better... Attitudes toward TIC include those that help realize and assimilate the TIC approach.

3.  page 2 line 89.  therapist = therapists

4.  page 3 line 143. After the Cieslak et al. reference, you need a period before It.

5.  page 3 line 146. Therefore, there is importance to explore = Therefore, it is important to explore....

6.  page 4 line 168. aged = ages

Round 2

Reviewer 1 Report

Comments and Suggestions for Authors

Thanks to the authors for the revision. I believe that the manuscript can be recommended for publication.